# Improving Stain Invariance of CNNs for Segmentation by Fusing Channel Attention and Domain-Adversarial Training

**Kudaibergen Abutalip**[1]                                      KUDAIBERGEN.ABUTALIP@MBZUAI.AC.AE

**Numan Saeed**[1]                                                      NUMAN.SAEED@MBZUAI.AC.AE

**Mustaqeem Khan**[1]                                          MUSTAQEEM.KHAN@MBZUAI.AC.AE

**Abdulmotaleb El Saddik**[1,2]                                      ELSADDIK@UOTTAWA.CA

[1] *Computer Vision Department, Mohamed Bin Zayed University of Artificial Intelligence, Abu Dhabi, UAE*

[2] *School of Electrical Engineering and Computer Science, University of Ottawa, Ottawa, Canada*

**Editors:** Accepted for publication at MIDL 2023

## Abstract

Variability in staining protocols, such as different slide preparation techniques, chemicals, and scanner configurations, can result in a diverse set of whole slide images (WSIs). This distribution shift can negatively impact the performance of deep learning models on unseen samples, presenting a significant challenge for developing new computational pathology applications. In this study, we propose a method for improving the generalizability of convolutional neural networks (CNNs) to stain changes in a single-source setting for semantic segmentation. Recent studies indicate that style features mainly exist as covariances in earlier network layers. We design a channel attention mechanism based on these findings that detects stain-specific features and modify the previously proposed stain-invariant training scheme. We reweigh the outputs of earlier layers and pass them to the stain-adversarial training branch. We evaluate our method on multi-center, multi-stain datasets and demonstrate its effectiveness through interpretability analysis. Our approach achieves substantial improvements over baselines and competitive performance compared to other methods, as measured by various evaluation metrics. We also show that combining our method with stain augmentation leads to mutually beneficial results and outperforms other techniques. Overall, our study makes significant contributions to the field of computational pathology.

**Keywords:** medical image segmentation, computational pathology, invariance, CNNs

## 1. Introduction

Transitioning to digital pathology (DP) can bring many practical benefits (Baxi et al., 2021), such as automation of time-costly manual tasks that are prone to human error (Barisoni et al., 2020; Neltner et al., 2012). However, many challenges of integrating artificial intelligence (AI) into DP workflow exist, and one of them is inter- and intra-institutional differences in staining protocols (Schömig-Markiefka et al., 2021). A significant shift in distribution might negatively affect the performance of deep learning (DL) models on new samples (Tellez et al., 2018). Recent studies mainly focus on image classification. In this work, we take a closer look at improving the generalization performance of CNNs on segmentation when data is available from one source and there is no access to a test set from a different source. We focus on the segmentation of different human tissue types that are important for further quantitative and qualitative analysis, disease diagnosis (Kannan et al.,

2019), and developing a fundamental understanding of how these biological structures are organized and interact with each other (Godwin et al., 2021).

Despite the growing number of research articles on adapting vision transformers (ViT) for medical applications, we use CNNs because recent works (Liu et al., 2022; Pinto et al., 2022, 2021; Bai et al., 2021; Lee et al., 2022) show that their full potential is still under-explored. They demonstrate that CNNs can perform better or on par with ViTs in terms of overall accuracy, generalization, and robustness to adversarial attacks. CNNs also have many advantages, such as less memory consumption and higher image throughput, which are crucial for wide-scale deployment.

The main contributions of this study are: (1) We propose a novel method for improving the generalization of CNNs based on detecting sensitive covariances and stain-invariant training. (2) Our method can be integrated with existing models, including pre-trained versions. (3) We show that combining our approach with stain augmentation mutually benefits each method. (4) An interpretability analysis of how the proposed method affects model parameters. Our implementation is available at github.com/katalip/ca-stinv-cnn.

## 2. Related Work

The digitization process involves several steps (Grizzle, 2009; Zarella et al., 2018), and different factors affect the final appearance of a WSI. For example, tissue extraction method, slice thickness, stain types, time spent in a reagent, and scanner setup can vary greatly. One of the pioneering methods used for reducing appearance discrepancies between WSIs is stain normalization (Macenko et al., 2009). To bring histopathology images into a common space, the authors propose to find optimal stain vectors, which correspond to each stain present in an image and form the basis for every pixel, by using singular value decomposition. Subsequent works (Sethi et al., 2016; Tellez et al., 2019) use this normalization method to improve DL models' generalization capability. They normalize training images using stain vectors of a reference image that is usually a test set sample. Other commonly used normalization techniques have been proposed by Vahadane et al. (2016) and Reinhard et al. (2001). The first limitation of this approach is accessibility (Ke et al., 2021) of a test set; secondly, potentially incomplete representation of distribution shift. Another group of works focuses on data augmentation. Tellez et al. (2018) have proposed a strategy based on color deconvolution for images stained with hematoxylin and eosin (H&E). Conventional approaches, such as HSV or color augmentation, also can be used but may generate unrealistic images. More recently, Chang et al. (2021) have proposed a Stain Mix-Up. This augmentation framework requires the computation of stain color matrices and stain density maps from source and target domain data using sparse non-negative matrix factorization (SNMF). RandStainNA is another recent development by Shen et al. (2022). The authors aim to address issues that both stain normalization and augmentation techniques have. Distant from data manipulation techniques, domain-adversarial (DA) training (Ganin et al., 2016) focuses on guiding a model to learn domain-invariant features. Some studies (Lafarge et al., 2019; Otálora et al., 2019) train custom DACNNs for mitosis classification, Gleason grading in prostate cancer, and nuclei segmentation tasks. DACNN achieves a much higher F1 score than the baseline method on classification; however, it shows a similar base performance on segmentation. Marini et al. (2021) develops the idea further and proposes using

optimal stain vectors, an intermediate step of Macenko normalization, instead of domain labels. The authors argue that such change guides a model to capture stain-invariant features. They extensively evaluate their colon and prostate tissue classification method and conclude that it is superior to its original version with domain labels, stain normalization, stainGAN (Shaban et al., 2019), color, and stain augmentation.

## 3. Methods

We propose a method for improving the stain generalization of a model for segmentation tasks in a single-source setting without access to the test set. Based on findings from Pan et al. (2018), we assume that features sensitive to stain changes mainly exist in earlier layers and use their intermediate outputs. We design a channel attention mechanism derived from detecting style-related covariances (Choi et al., 2021) to focus on domain-specific features. Reweighed feature maps are passed to the stain-invariant training branch that guides a model to capture more robust representations. We describe each step of the overall pipeline (Figure 1) in detail below.

### 3.1. Detecting Stain-Specific Features

Studies (Gatys et al., 2015, 2016) report that style information is encoded as feature covariances. We aim to use this information in the form of channel attention before passing feature maps to the adversarial training branch. With this step, we expect to suppress only stain-specific features selectively. Let $\mathbf{F} \in \mathbb{R}^{C \times HW}$ denote an intermediate feature map, where $C, H, W$ represent channel, height, and width dimensions accordingly. The covariance matrix is computed as follows

$$\mathbf{\Sigma}(\mathbf{F}) = \frac{1}{HW}(\mathbf{F})(\mathbf{F})^\top \in \mathbb{R}^{C \times C} \tag{1}$$

To detect sensitive covariances, we first apply stain augmentation (perturbation of stain vectors) to the given input image and compute the second covariance matrix $\mathbf{\Sigma}(\mathbf{F}') \in \mathbb{R}^{C \times C}$ of its intermediate feature representation $\mathbf{F}' \in \mathbb{R}^{C \times HW}$ obtained from the same encoder layer. Next, a variance matrix $\mathbf{V} \in \mathbb{R}^{C \times C}$ that represents the difference between covariance matrices of the input image and its augmented version is computed as

$$\boldsymbol{\mu_\sigma} = \frac{1}{2}(\mathbf{\Sigma}(\mathbf{F}) + \mathbf{\Sigma}(\mathbf{F}')) \tag{2}$$

$$\mathbf{V} = \frac{1}{2}((\mathbf{\Sigma}(\mathbf{F}) - \boldsymbol{\mu_\sigma})^2 + (\mathbf{\Sigma}(\mathbf{F}') - \boldsymbol{\mu_\sigma})^2) \tag{3}$$

Then we linearly map variance values to channel weights using a fully connected layer $\mathbf{M} \in \mathbb{R}^{C \times 1}$, followed by a sigmoid activation function. For multiple images, we preserve batch dimension during computations.

### 3.2. Stain-Invariant Training

After reweighing feature maps $\mathbf{F} \in \mathbb{R}^{C \times H \times W}$ across channel dimension, they are passed to the network branch that enforces invariance of the model to stain changes.

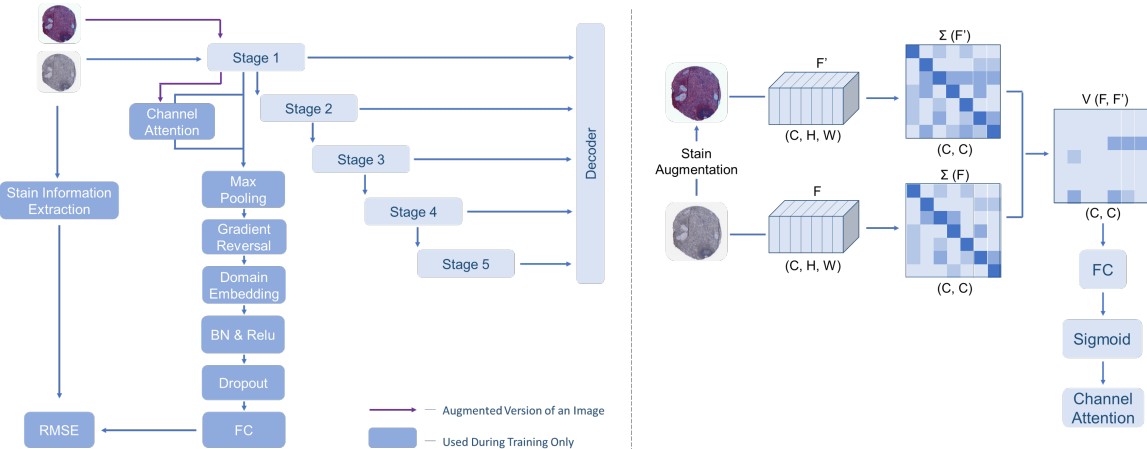

Figure 1: Overall pipeline of the proposed method. An augmented version of an input image is used for detecting sensitive covariances. Then we compute channel attention from the variance matrix ($\mathbf{V}$) that represents the difference between covariance matrices ($\mathbf{\Sigma}$) of the input image and its transformed version. Next, reweighed feature maps are passed to the stain-invariant training branch.

The key idea is based on domain-adversarial training (Ganin et al., 2016), but instead of predicting domain labels, Marini et al. (2021) proposed to use optimal stain vectors. If we consider two optimal stain vectors, then each row of the matrix $\mathbf{S} \in \mathbb{R}^{3\times 2}$ contains corresponding $R$, $G$, $B$ intensities of each vector.

In our setting, the first components of this branch are the downsampling layer, the gradient reversal layer (GRL) that acts as an identity mapping during the forward pass and negates gradients during the backward pass. The GRL forces feature representations over different distributions to be similar. The next layers include embedding, batch normalization, non-linear activation, and dropout. Flattened outputs of the prediction head $\hat{\mathbf{S}} \in \mathbb{R}^{3\times 2}$ are used for computing RMSE loss

$$\mathcal{L}_s = \sqrt{\frac{1}{N}\Sigma_{i=1}^{N}(\mathbf{S}_i - \hat{\mathbf{S}}_i)^2} \tag{4}$$

$$\mathcal{L} = \mathcal{L}_{task} + \alpha * \mathcal{L}_s \tag{5}$$

where $\mathcal{L}$ is a total loss, $\mathcal{L}_{task}$ in this case represents loss for segmentation, and $\alpha$ is a hyperparameter. In this scenario, a network is expected to learn generalizable representations that are more robust to unseen stain styles.

According to findings from Pan et al. (2018), we make simple yet effective modifications to the method. By considering that style information is mostly present in earlier layers in the form of feature covariances, we propose to use the outputs of the first stage for stain-adversarial training. In later sections, we empirically show the importance of positioning and that improper placement might even negatively affect model performance on segmentation.

Table 1: Comparison of Dice scores with ResNet-50 as encoder. The HPA dataset is used for training. STAUG: Stain augmentation. S: Strong. L: Light. P: Photometric.

| Method | Validation: HPA | | | | Public Test HPA+HuBMAP | Private Test HuBMAP | NEPTUNE | | | | HuBMAP21 | AIDPATH |
|---|---|---|---|---|---|---|---|---|---|---|---|---|
| | - | H&E | PAS | TRI | | H&E + PAS | H&E | PAS | TRI | SIL | PAS | PAS |
| Baseline | 0.691 ± 0.044 | 0.605 ± 0.053 | 0.509 ± 0.066 | 0.493 ± 0.098 | 0.477 ± 0.064 | 0.399 ± 0.079 | 0.490 ± 0.036 | 0.542 ± 0.028 | 0.518 ± 0.048 | 0.386 ± 0.030 | 0.249 ± 0.075 | 0.363 ± 0.039 |
| HSV | 0.729 ± 0.012 | 0.690 ± 0.017 | 0.586 ± 0.031 | 0.648 ± 0.035 | 0.616 ± 0.012 | 0.554 ± 0.017 | 0.668 ± 0.057 | 0.665 ± 0.057 | 0.666 ± 0.071 | 0.549 ± 0.147 | 0.386 ± 0.045 | 0.467 ± 0.032 |
| STAUG-S | 0.734 ± 0.018 | **0.695** ± **0.023** | 0.602 ± 0.065 | **0.664** ± **0.040** | 0.610 ± 0.013 | 0.550 ± 0.028 | **0.738** ± **0.088** | 0.727 ± 0.119 | 0.759 ± 0.069 | 0.614 ± 0.127 | 0.436 ± 0.029 | 0.453 ± 0.069 |
| RandStainNA | 0.566 ± 0.017 | 0.489 ± 0.020 | 0.441 ± 0.039 | 0.432 ± 0.024 | 0.291 ± 0.014 | 0.220 ± 0.041 | 0.384 ± 0.089 | 0.583 ± 0.035 | 0.392 ± 0.048 | 0.461 ± 0.079 | 0.156 ± 0.088 | 0.352 ± 0.006 |
| STAUG-L | **0.736** ± **0.010** | 0.661 ± 0.010 | **0.618** ± **0.024** | 0.622 ± 0.024 | 0.560 ± 0.023 | 0.484 ± 0.026 | 0.634 ± 0.024 | 0.693 ± 0.017 | 0.632 ± 0.009 | 0.522 ± 0.108 | 0.437 ± 0.062 | 0.466 ± 0.038 |
| ISW-P | 0.715 ± 0.010 | 0.605 ± 0.023 | 0.474 ± 0.029 | 0.579 ± 0.050 | 0.440 ± 0.016 | 0.360 ± 0.024 | 0.460 ± 0.056 | 0.558 ± 0.014 | 0.483 ± 0.090 | 0.428 ± 0.074 | 0.237 ± 0.045 | 0.397 ± 0.028 |
| ISW-STAUG | 0.730 ± 0.016 | 0.606 ± 0.026 | 0.454 ± 0.081 | 0.582 ± 0.021 | 0.467 ± 0.065 | 0.388 ± 0.069 | 0.375 ± 0.161 | 0.500 ± 0.122 | 0.457 ± 0.100 | 0.314 ± 0.096 | 0.233 ± 0.036 | 0.283 ± 0.105 |
| Proposed | 0.721 ± 0.025 | 0.631 ± 0.018 | 0.567 ± 0.014 | 0.568 ± 0.071 | 0.526 ± 0.059 | 0.453 ± 0.073 | 0.593 ± 0.025 | 0.637 ± 0.057 | 0.566 ± 0.130 | 0.501 ± 0.115 | 0.295 ± 0.036 | **0.476 ± 0.030** |
| Proposed + STAUG-S | 0.711 ± 0.020 | 0.693 ± 0.025 | 0.616 ± 0.023 | 0.650 ± 0.025 | **0.622 ± 0.029** | **0.567 ± 0.029** | 0.736 ± 0.060 | **0.786 ± 0.028** | **0.808 ± 0.023** | **0.733 ± 0.040** | **0.483 ± 0.018** | 0.470 ± 0.051 |

## 4. Experiments and Results

### 4.1. Datasets

We use four public datasets for conducting experiments. **HPA + HuBMAP 2022** (Howard et al., 2022) contains WSIs of tissues from five organs: glomerulus (kidney), white pulp (spleen), alveolus (lung), glandular acinus (prostate), colonic crypt (large intestine). The task is to segment a superclass of functional tissue units against a background. There are 351 images from HPA stained with antibodies visualized with 3,3'-diaminobenzidine (DAB) and counterstained with hematoxylin in the training set, 81 and 351 samples from both sources in the public test, and 238 WSIs from HuBMAP stained H&E or periodic acid-Schiff (PAS) in the private test set. There was no access to both test sets during this study, and results were obtained via online submissions. A subset of **NEPTUNE** (Jayapandian et al., 2021) is comprised of glomerulus biopsy slides, stained with H&E (81), PAS (203), periodic acid–methenamine silver (SIL, 123), and Masson trichrome (TRI, 137). **AIDPATH** (Bueno et al., 2020) and **HuBMAP21 Kidney** (Godwin et al., 2021) also contain 31 and 20 large (e.g. 31299×44066) glomerulus samples (PAS) respectively. We resize training images to 768×768 resolution and use test samples at resolutions that match the pixel size of the HPA train set. Details related to magnification, slice thickness differences between the datasets, and example illustrations are available in Appendix A.

### 4.2. Baseline and Compared Methods

Baseline is a 2D U-Net architecture (Ronneberger et al., 2015) with ResNet-50 (He et al., 2016), or ConvNeXt-Tiny (Liu et al., 2022) as an encoder. Similar to (Shen et al., 2022; Chang et al., 2021; Marini et al., 2021), we compare the proposed method with approaches that specifically aim to improve the stain generalization capability of the model and do not require access to the test data. Augmentation in HSV color space is a common method for

Table 2: Comparison of Dice scores with ConvNext-Tiny as an encoder. The HPA dataset is used for training. STAUG: Stain augmentation. S: Strong. L: Light.

| Method | Validation: HPA | | | | Public Test HPA+HuBMAP | Private Test HuBMAP | NEPTUNE | | | | HuBMAP21 | AIDPATH |
|---|---|---|---|---|---|---|---|---|---|---|---|---|
| | - | H&E | PAS | TRI | | H&E + PAS | H&E | PAS | TRI | SIL | PAS | PAS |
| Baseline | **0.750** ± **0.001** | **0.712** ± **0.006** | 0.636 ± 0.023 | 0.688 ± 0.011 | 0.585 ± 0.027 | 0.511 ± 0.034 | 0.760 ± 0.055 | 0.656 ± 0.080 | 0.750 ± 0.027 | 0.531 ± 0.078 | 0.515 ± 0.068 | 0.408 ± 0.111 |
| HSV | 0.732 ± 0.017 | 0.711 ± 0.029 | 0.640 ± 0.029 | **0.699** ± **0.029** | 0.593 ± 0.012 | 0.531 ± 0.012 | 0.812 ± 0.027 | 0.697 ± 0.041 | **0.828** ± **0.015** | 0.612 ± 0.043 | 0.517 ± 0.019 | 0.458 ± 0.018 |
| STAUG-S | 0.735 ± 0.011 | 0.698 ± 0.010 | 0.658 ± 0.016 | 0.685 ± 0.013 | 0.588 ± 0.033 | 0.528 ± 0.035 | 0.787 ± 0.029 | 0.696 ± 0.057 | 0.790 ± 0.025 | 0.575 ± 0.095 | 0.488 ± 0.035 | 0.402 ± 0.038 |
| RandStainNA | 0.679 ± 0.009 | 0.668 ± 0.017 | **0.660** ± **0.015** | 0.644 ± 0.015 | 0.563 ± 0.015 | 0.512 ± 0.021 | 0.773 ± 0.012 | **0.771** ± **0.008** | 0.647 ± 0.027 | 0.598 ± 0.016 | 0.497 ± 0.009 | 0.467 ± 0.038 |
| STAUG-L | 0.736 ± 0.014 | 0.711 ± 0.022 | 0.639 ± 0.022 | 0.690 ± 0.027 | 0.585 ± 0.013 | 0.509 ± 0.016 | 0.756 ± 0.069 | 0.664 ± 0.052 | 0.797 ± 0.005 | 0.603 ± 0.018 | 0.493 ± 0.025 | 0.391 ± 0.055 |
| ISW-P | 0.735 ± 0.022 | 0.670 ± 0.060 | 0.592 ± 0.074 | 0.661 ± 0.045 | 0.553 ± 0.044 | 0.477 ± 0.041 | 0.752 ± 0.040 | 0.618 ± 0.051 | 0.768 ± 0.019 | 0.418 ± 0.033 | 0.237 ± 0.045 | 0.362 ± 0.059 |
| ISW-STAUG | 0.733 ± 0.016 | 0.685 ± 0.025 | 0.616 ± 0.012 | 0.657 ± 0.026 | 0.599 ± 0.006 | 0.525 ± 0.006 | 0.733 ± 0.054 | 0.628 ± 0.058 | 0.753 ± 0.010 | 0.471 ± 0.061 | 0.498 ± 0.050 | 0.385 ± 0.056 |
| Proposed | 0.737 ± 0.011 | 0.708 ± 0.024 | 0.624 ± 0.006 | 0.685 ± 0.016 | 0.600 ± 0.012 | 0.533 ± 0.018 | 0.788 ± 0.018 | 0.704 ± 0.026 | 0.785 ± 0.007 | 0.598 ± 0.060 | 0.497 ± 0.069 | **0.468** ± **0.033** |
| Proposed + STAUG-S | 0.738 ± 0.008 | 0.705 ± 0.003 | 0.639 ± 0.007 | 0.694 ± 0.008 | **0.622 ± 0.016** | **0.562 ± 0.013** | **0.815** ± **0.036** | 0.745 ± 0.037 | 0.825 ± 0.023 | **0.621** ± **0.066** | **0.555 ± 0.023** | 0.462 ± 0.020 |

increasing stain diversity in a train set. RandStainNA (Shen et al., 2022) is a recent augmentation framework that efficiently fuses stain normalization with augmentation. For light stain augmentation (Marini et al., 2021), we linearly scale and shift values of optimal stain vectors by a small amount. For a stronger version, we make element-wise perturbations of each vector. According to our observations, this generates more clinically-questionable samples. We also try to adapt instance-selective whitening loss (ISW) from RobustNet (Choi et al., 2021), a domain-generalization method for the semantic segmentation of natural images. It is important to mention that we do not use the full network, which has some architectural modifications but adapt the loss for the encoders. ISW requires a second augmented version of an image, and for that, we use originally proposed photometric transformations and experiment with stain augmentation.

### 4.3. Metrics and Evaluation

We use the Dice score as the main metric and report precision and recall values in Appendix C. Two backbone networks are used for evaluation. We perform three runs for each experiment and report mean and standard deviation values. The training part of the HPA + HuBMAP 2022 dataset is used for training (263) and validation (83), while other datasets from external sources are used for testing. Additionally, we use stain normalization for synthesizing differently stained versions of the validation set. We use small batch size (Hoffer et al., 2017), additional augmentations, and an appropriate number of iterations for each encoder to prevent models from overfitting. Implementation details are provided in Appendix B.

### 4.4. Generalization Performance

We first outline results with ResNet-50 as an encoder (Table 1). The proposed method improves the generalization performance of the baseline across all test datasets and valida-

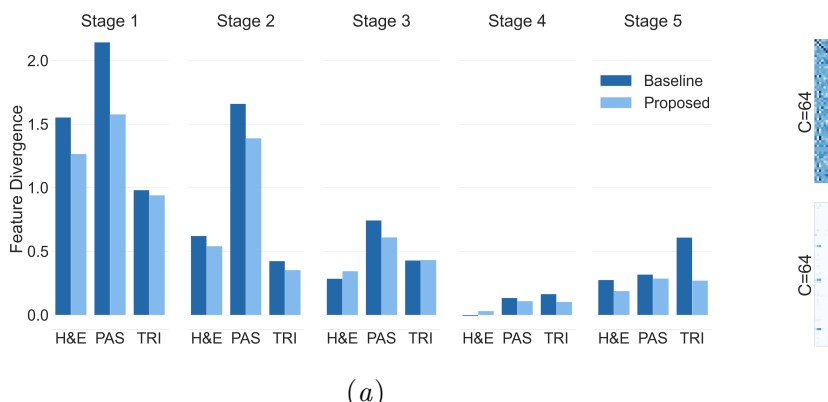
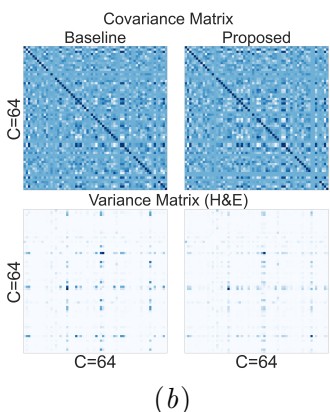

Figure 2: (*a*) Feature divergence analysis of the baseline and our method on stain normalized (H&E, PAS, TRI) validation set. (*b*) Visualization of covariance and variance matrices (H&E).

tion. It also obtains better results compared to RandStainNA and ISW variations. Though stain and HSV augmentations seem to be more effective, it is essential to note that our method aims to maximize the generalization capability of the baseline by affecting model parameters. In contrast, the aforementioned augmentations are closer to using more data. Considering the random nature of augmentations, it is hard to simulate all possible stain shifts in practice. We think having an explicit way of regularizing a model can be reassuring during the deployment phase. Moreover, the results show that combining stain augmentation with the proposed method mutually benefits each approach. Former makes a model less biased toward unrealistic stain styles usually generated by stain augmentation. Increasing stain diversity benefits our method by incorporating information from new domains. The combination achieves the maximum performance compared to all methods on test datasets.

ConvNeXt-Tiny converges much faster compared to its predecessor. The proposed method increases the generalization capacity of the baseline on all test sets (Table 2), except HuBMAP21 Kidney. It also achieves a larger performance increase on public and private test sets than other approaches. Stain and HSV augmentations also positively affect the model's generalizability, but to a lesser extent than the ResNet-50. We observe that complementary usage of the proposed method and stain augmentation outperform other methods. We outline precision, recall scores, and further qualitative analysis in Appendix C.

## 4.5. Interpretability Analysis

We perform additional analysis with ResNet-50 to understand better why modifying the training process helps to attain better generalization. Similar to (Pan et al., 2018), we compute KL divergence between the original validation set and its normalized stain versions (H&E, PAS, TRI). Let $\mathbf{F}$ denote a feature map, and $\mu$, $\sigma^2$ denote its mean and variance values, respectively. Then symmetric KL divergence between two distributions $S$ and $S'$ is

computed as

$$D(\mathbf{F_S}||\mathbf{F_{S'}}) = KL(\mathbf{F_S}||\mathbf{F_{S'}}) + KL(\mathbf{F_{S'}}||\mathbf{F_S}) \tag{6}$$

$$KL(\mathbf{F_S}||\mathbf{F_{S'}}) = \log\frac{\sigma_{S'}}{\sigma_S} + \frac{\sigma_S^2 + (\mu_S - \mu_{S'})^2}{2\sigma_{S'}^2} - \frac{1}{2} \tag{7}$$

We use the average divergence of all channels for each image and report the mean value for the whole set. From Figure $2(a)$, it can be seen that there is less feature divergence for the modified network, suggesting that learned representations are more generalizable. Less discrepancy between different distributions can be observed at later stages, the same pattern authors of IBNnet (Pan et al., 2018) have observed.

Next, we visualize mean covariance and variance matrices (Figure $2(b)$) and observe that the covariance matrix computed from intermediate outputs of the modified model has more bright spots. This might result from suppressing sensitive covariances previously present in the baseline. There are also fewer activations in the variance matrix for H&E, which means that the distribution

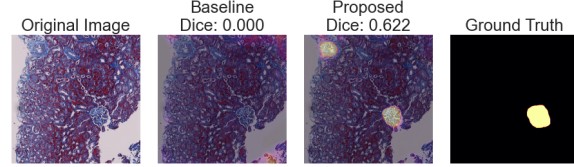

Figure 3: Segmentation results on unseen stain style (TRI, NEPTUNE).

shift is causing less divergence for all channels on average. The same pattern can be observed for TRI and PAS variance matrices (Figure 10 in Appendix C), however, for the latter, some of the brighter spots appear darker compared to the baseline.

t-SNE visualization (Figure 9 in Appendix C) of learned representations from stain normalized and original image versions of the first and last encoder stages show that they are closer when we integrate our method into training scheme. Though it is harder to notice this with stage one outputs, fifth-stage visualization makes it more apparent. We can also see that affecting the earlier layer impacts the later stages.

As seen in Figure 11 and Figure 12 in Appendix C, the baseline model exhibits more performance drop when stain change occurs for the given image. In the case of Figure 3, it completely misses the glomerulus present in the image, while the proposed modification helps to alleviate the problem. We think in a clinical setting, where every image contains valuable information, such omission can negatively affect the overall patient treatment process. Though quantitative results, on average, are the main indicator of the effectiveness of the proposed and similar methods, analysis of individual use cases can help facilitate better understanding.

We also investigate how using the proposed method after later encoder stages, different design choices, and the effect of channel attention with ResNet-50 in Appendix D.

## 5. Conclusion

In this work, we propose a method for enforcing the invariance of CNNs to stain changes for segmentation. The method comprises detecting sensitive covariances and subsequent stain-invariant training branch. We demonstrate the effectiveness of the presented method's separate and combined usage. However, we also find some limitations, such as sensitivity to design choices and a requirement of additional adaptation to other backbones and datasets.

In this study, we hypothesize that some covariances might still be helpful for certain objects, such as white pulp in the spleen. Future works might focus on filtering rather than suppressing all of them. Another promising direction is investigating whether our approach can be integrated into memory-efficient training pipelines for WSIs. For example, a recent study (Zhang et al., 2022) showed that models can be trained on full-resolution images without tiling or resizing via locally supervised learning. Our method may be suitable for such a use-case scenario. We hope that outlined findings will open new research directions for domain generalization with histological data.

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

## Appendix A. Dataset Details

WSI's from HPA have an image size of 3000×3000, tissue thickness of 4 µm, and pixel size of 0.4 µm. Image sizes from HuBMAP vary from 4500×4500 to 160×160. Tissue thicknesses are different for each organ: 10 µm for the kidney, 8 µm for the large intestine, 4 µm for the spleen, 5 µm for the lung, and 5 µm for the prostate. The same goes for pixel sizes: 0.5 µm for the kidney, 0.2290 µm for the large intestine, 0.7562 µm for the lung, 0.4945 µm for

the spleen, and 6.263 µm for the prostate. Lung WSIs contain collapsed, uncollapsed views of alveolus cut horizontally or vertically (Figure 6). Some studies (Chlipala et al., 2021; Yagi and Gilbertson, 2008) provide analysis on how different values of slice thickness affect overall image quality and color intensity. All images from the NEPTUNE dataset have a resolution of 3000×3000, and a pixel size of 0.24 µm (5× magnification). AIDPATH WSI's range between 21651×10498 pixels and 49799×32359 pixels acquired at 20x magnification. The slice thickness is 4 µm. We use a publicly available tiled (1024×1024) version. For the HuBMAP21 Kidney, a tiled (512×512) version is used in this study. Example images can be viewed on Figure 5 and Figure 7. An example of a synthesized validation image can be seen on Figure 8.

Similar to (Marini et al., 2021), we check whether stain-invariant training can be applied to this dataset by examining intra-stain variability (Figure 4). According to Reina et al. (2020), using full WSIs instead of tiling gives better and more stable results for semantic segmentation. The authors recommend using tiling only when memory constraints can not be met.

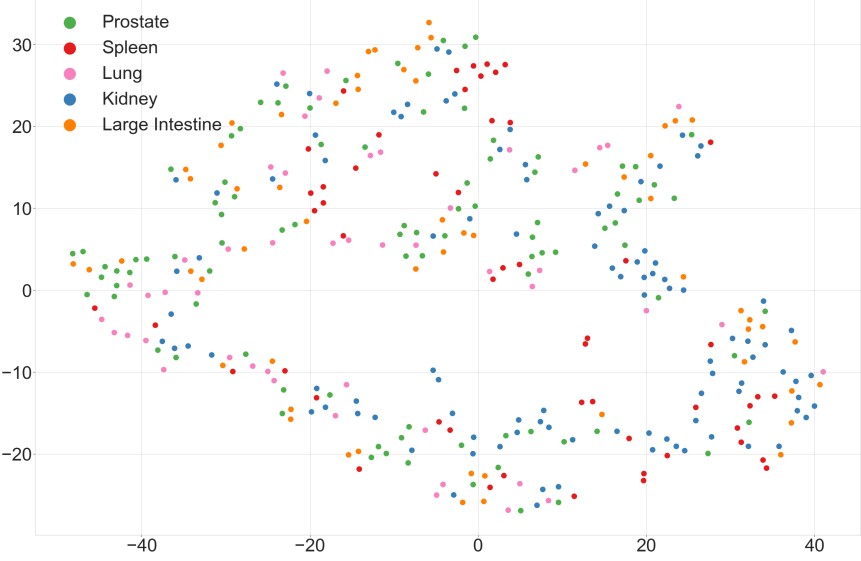

Figure 4: Stain variability present in the train set. Optimal stain vectors are projected to lower dimensional space with t-SNE.

## Appendix B. Implementation Details

We use an AdamW optimizer with a learning rate of $5e-5$, weight decay of $1e-3$, and batch size of 4. Experiments are run for 80 and 40 epochs for ResNet-50 and ConvNeXt-Tiny respectively on NVIDIA Quadro RTX 6000 24 GB GPU. Base augmentations included random horizontal, and vertical flips, rotation, shifting, and scaling within the range of [-0.2, 0.2], and random brightness, and contrast changes within the range of [-0.2, 0.2]. For

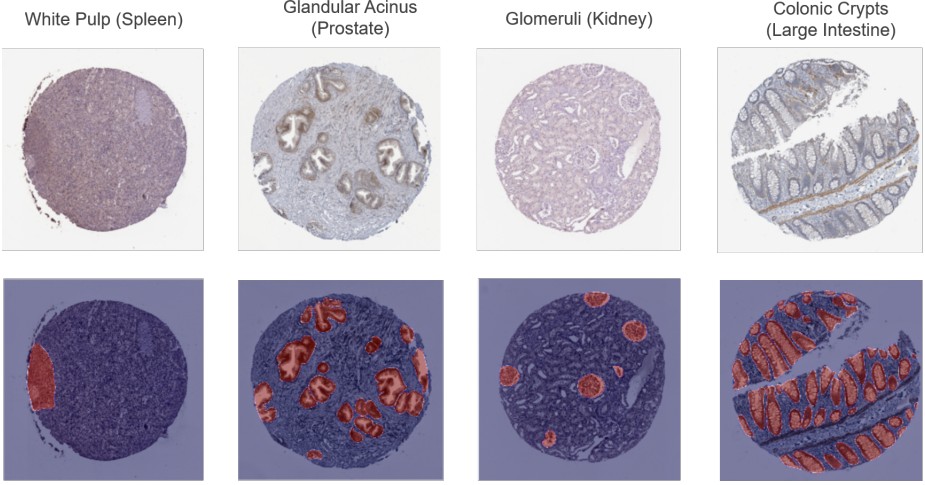

Figure 5: Examples from HPA+HuBMAP 2022 dataset.

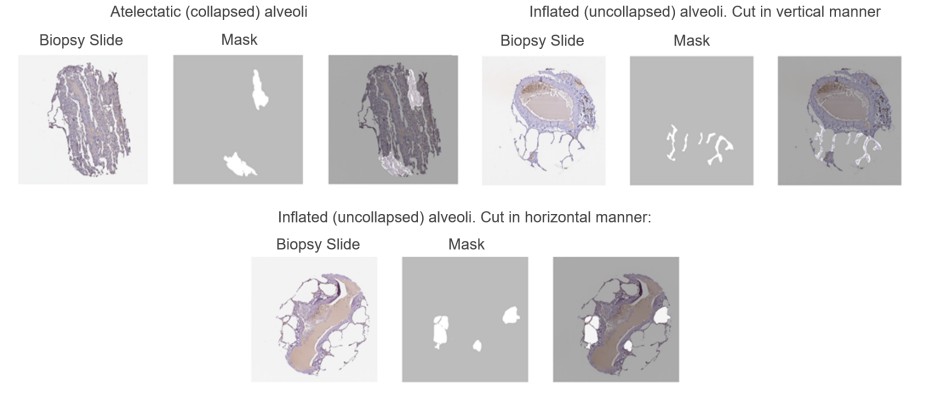

Figure 6: Alveolus (lung) examples from HPA+HuBMAP 2022 dataset.

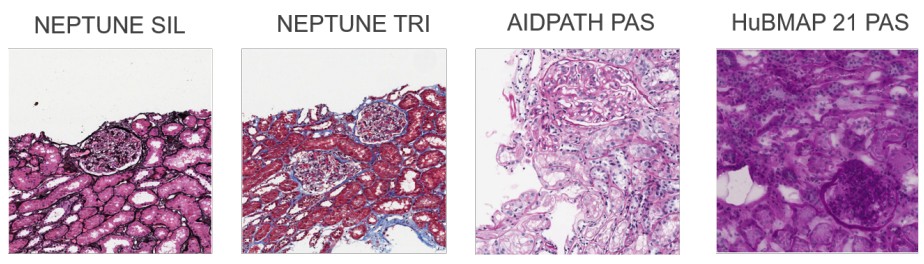

Figure 7: Examples from other test datasets.



Figure 8: Example of synthesized validation image.

ISW, original transformations are color jittering and gaussian blur, and we try to adapt it by using strong stain augmentation. Encoder stages refer to model stages. In the case of ResNet-50, we use outputs of the first convolutional layer (conv1) after batch normalization and non-linear activation. For ConvNext-Tiny, we use outputs of the first (zeroth index) downsampling layer. We set $\alpha$ to 0.5 in the combined loss.

## Appendix C. Extended Results

### C.1. Precision and Recall Scores

Table 3: Precision scores with ResNet-50 as encoder. STAUG: Stain augmentation. S: Strong. L: Light. P: Photometric.

| Method | Validation: HPA | | | | NEPTUNE | | | | HuBMAP21 | AIDPATH |
|---|---|---|---|---|---|---|---|---|---|---|
| | - | H&E | PAS | TRI | H&E | PAS | TRI | SIL | PAS | PAS |
| Baseline | 0.754 ± 0.028 | 0.724 ± 0.038 | 0.646 ± 0.042 | 0.585 ± 0.060 | 0.655 ± 0.076 | 0.571 ± 0.089 | 0.573 ± 0.047 | 0.417 ± 0.092 | 0.249 ± 0.075 | 0.453 ± 0.058 |
| HSV | 0.777 ± 0.016 | 0.738 ± 0.036 | 0.612 ± 0.014 | 0.705 ± 0.042 | 0.817 ± 0.071 | 0.707 ± 0.068 | 0.675 ± 0.071 | 0.597 ± 0.142 | 0.381 ± 0.055 | 0.557 ± 0.037 |
| STAUG-S | 0.787 ± 0.014 | **0.758** ± **0.014** | **0.687** ± **0.060** | 0.740 ± 0.026 | **0.887** ± **0.038** | 0.805 ± 0.088 | 0.782 ± 0.053 | 0.664 ± 0.059 | 0.483 ± 0.067 | 0.566 ± 0.051 |
| RandStainNA | 0.646 ± 0.038 | 0.587 ± 0.068 | 0.588 ± 0.046 | 0.557 ± 0.039 | 0.609 ± 0.050 | 0.734 ± 0.048 | 0.419 ± 0.084 | 0.542 ± 0.106 | 0.250 ± 0.132 | 0.476 ± 0.035 |
| STAUG-L | 0.775 ± 0.012 | 0.696 ± 0.033 | 0.650 ± 0.005 | 0.672 ± 0.037 | 0.741 ± 0.052 | 0.750 ± 0.052 | 0.617 ± 0.010 | 0.566 ± 0.140 | 0.456 ± 0.034 | 0.559 ± 0.018 |
| ISW-P | 0.776 ± 0.009 | 0.624 ± 0.048 | 0.496 ± 0.075 | 0.573 ± 0.065 | 0.498 ± 0.083 | 0.569 ± 0.061 | 0.426 ± 0.114 | 0.443 ± 0.083 | 0.196 ± 0.071 | 0.446 ± 0.035 |
| ISW-STAUG | 0.779 ± 0.012 | 0.671 ± 0.038 | 0.523 ± 0.094 | 0.626 ± 0.082 | 0.469 ± 0.143 | 0.548 ± 0.088 | 0.492 ± 0.140 | 0.323 ± 0.126 | 0.204 ± 0.011 | 0.379 ± 0.101 |
| Proposed | **0.789** ± **0.010** | 0.675 ± 0.071 | 0.614 ± 0.080 | 0.600 ± 0.089 | 0.705 ± 0.082 | 0.624 ± 0.131 | 0.528 ± 0.216 | 0.485 ± 0.217 | 0.258 ± 0.087 | 0.547 ± 0.025 |
| Proposed + STAUG-S | 0.779 ± 0.005 | 0.757 ± 0.012 | 0.673 ± 0.023 | **0.758** ± **0.019** | 0.876 ± 0.039 | **0.841** ± **0.061** | **0.846** ± **0.009** | **0.804** ± **0.050** | **0.549 ± 0.018** | **0.567 ± 0.022** |

Table 4: Recall scores with ResNet-50 as encoder. STAUG: Stain augmentation. S: Strong. L: Light. P: Photometric.

| Method | Validation: HPA | | | | NEPTUNE | | | | HuBMAP21 | AIDPATH |
|---|---|---|---|---|---|---|---|---|---|---|
| | - | H&E | PAS | TRI | H&E | PAS | TRI | SIL | PAS | PAS |
| Baseline | 0.684 ± 0.042 | 0.595 ± 0.036 | 0.498 ± 0.092 | 0.508 ± 0.118 | 0.445 ± 0.044 | 0.627 ± 0.099 | 0.563 ± 0.065 | 0.492 ± 0.076 | 0.369 ± 0.108 | 0.398 ± 0.118 |
| HSV | 0.731 ± 0.011 | 0.705 ± 0.044 | 0.665 ± 0.056 | 0.666 ± 0.047 | 0.604 ± 0.047 | 0.682 ± 0.047 | 0.724 ± 0.043 | 0.581 ± 0.131 | 0.561 ± 0.051 | 0.463 ± 0.036 |
| STAUG-S | 0.729 ± 0.029 | 0.684 ± 0.031 | 0.602 ± 0.084 | 0.648 ± 0.053 | 0.670 ± 0.101 | 0.714 ± 0.140 | 0.777 ± 0.082 | 0.655 ± 0.168 | 0.500 ± 0.044 | 0.429 ± 0.077 |
| RandStainNA | 0.562 ± 0.025 | 0.503 ± 0.047 | 0.399 ± 0.032 | 0.419 ± 0.052 | 0.318 ± 0.088 | 0.535 ± 0.047 | 0.473 ± 0.094 | 0.507 ± 0.133 | 0.135 ± 0.073 | 0.333 ± 0.032 |
| STAUG-L | **0.751 ± 0.012** | **0.722 ± 0.053** | **0.680 ± 0.048** | 0.667 ± 0.044 | 0.610 ± 0.031 | 0.698 ± 0.010 | 0.724 ± 0.036 | 0.577 ± 0.081 | 0.528 ± 0.069 | 0.461 ± 0.051 |
| ISW-P | 0.714 ± 0.011 | 0.696 ± 0.021 | 0.598 ± 0.074 | **0.701 ± 0.055** | 0.540 ± 0.036 | 0.653 ± 0.066 | 0.736 ± 0.076 | 0.572 ± 0.108 | **0.689 ± 0.129** | **0.474 ± 0.107** |
| ISW-STAUG | 0.731 ± 0.017 | 0.636 ± 0.069 | 0.539 ± 0.104 | 0.656 ± 0.080 | 0.429 ± 0.222 | 0.554 ± 0.162 | 0.569 ± 0.088 | 0.448 ± 0.071 | 0.589 ± 0.160 | 0.288 ± 0.122 |
| Proposed | 0.713 ± 0.047 | 0.703 ± 0.106 | 0.639 ± 0.095 | 0.669 ± 0.040 | 0.594 ± 0.091 | 0.778 ± 0.078 | 0.786 ± 0.057 | 0.721 ± 0.088 | 0.648 ± 0.178 | 0.508 ± 0.081 |
| Proposed + STAUG-S | 0.704 ± 0.041 | 0.693 ± 0.045 | 0.634 ± 0.058 | 0.627 ± 0.044 | **0.679 ± 0.093** | **0.781 ± 0.058** | **0.799 ± 0.036** | **0.727 ± 0.062** | 0.499 ± 0.060 | 0.457 ± 0.075 |

Table 5: Precision scores with ConvNext-Tiny as an encoder. STAUG: Stain augmentation. S: Strong. L: Light. P: Photometric.

| Method | Validation: HPA | | | | NEPTUNE | | | | HuBMAP21 | AIDPATH |
|---|---|---|---|---|---|---|---|---|---|---|
| | - | H&E | PAS | TRI | H&E | PAS | TRI | SIL | PAS | PAS |
| Baseline | 0.767 ± 0.002 | 0.750 ± 0.009 | 0.675 ± 0.037 | 0.725 ± 0.016 | 0.866 ± 0.028 | 0.693 ± 0.054 | 0.757 ± 0.009 | 0.486 ± 0.042 | 0.505 ± 0.093 | 0.493 ± 0.095 |
| HSV | 0.775 ± 0.005 | 0.767 ± 0.007 | 0.717 ± 0.017 | 0.756 ± 0.009 | 0.932 ± 0.022 | 0.717 ± 0.049 | 0.842 ± 0.017 | 0.573 ± 0.076 | 0.533 ± 0.069 | 0.543 ± 0.027 |
| STAUG-S | 0.737 ± 0.036 | **0.787 ± 0.055** | 0.691 ± 0.052 | 0.723 ± 0.037 | 0.893 ± 0.039 | 0.727 ± 0.102 | 0.766 ± 0.054 | 0.547 ± 0.128 | 0.515 ± 0.093 | 0.465 ± 0.071 |
| RandStainNA | 0.719 ± 0.018 | 0.667 ± 0.038 | 0.723 ± 0.018 | 0.679 ± 0.026 | 0.907 ± 0.003 | **0.835 ± 0.010** | 0.593 ± 0.056 | 0.529 ± 0.018 | 0.580 ± 0.042 | **0.582 ± 0.021** |
| STAUG-L | 0.775 ± 0.002 | 0.745 ± 0.017 | 0.688 ± 0.013 | **0.758 ± 0.004** | 0.905 ± 0.013 | 0.739 ± 0.032 | 0.810 ± 0.031 | 0.620 ± 0.074 | 0.531 ± 0.054 | 0.483 ± 0.053 |
| ISW-P | 0.771 ± 0.019 | 0.737 ± 0.019 | 0.677 ± 0.040 | 0.732 ± 0.012 | 0.893 ± 0.034 | 0.679 ± 0.032 | 0.796 ± 0.022 | 0.415 ± 0.109 | 0.497 ± 0.086 | 0.448 ± 0.037 |
| ISW-STAUG | 0.774 ± 0.003 | 0.744 ± 0.013 | 0.674 ± 0.018 | 0.727 ± 0.012 | 0.897 ± 0.024 | 0.705 ± 0.045 | 0.802 ± 0.041 | 0.509 ± 0.106 | 0.551 ± 0.062 | 0.478 ± 0.051 |
| Proposed | **0.784 ± 0.026** | 0.754 ± 0.035 | 0.704 ± 0.062 | 0.757 ± 0.025 | 0.899 ± 0.058 | 0.743 ± 0.055 | 0.798 ± 0.035 | 0.582 ± 0.066 | 0.496 ± 0.132 | 0.542 ± 0.017 |
| Proposed + STAUG-S | 0.755 ± 0.047 | 0.753 ± 0.028 | **0.728 ± 0.018** | 0.753 ± 0.037 | **0.941 ± 0.026** | 0.812 ± 0.054 | **0.847 ± 0.035** | **0.643 ± 0.093** | **0.601 ± 0.071** | 0.558 ± 0.010 |

Table 6: Recall scores with ConvNext-Tiny as an encoder. STAUG: Stain augmentation. S: Strong. L: Light. P: Photometric.

| Method | Validation: HPA | | | | NEPTUNE | | | | HuBMAP21 | AIDPATH |
|---|---|---|---|---|---|---|---|---|---|---|
| | - | H&E | PAS | TRI | H&E | PAS | TRI | SIL | PAS | PAS |
| Baseline | 0.771 ± 0.003 | 0.730 ± 0.008 | 0.647 ± 0.026 | 0.708 ± 0.010 | 0.708 ± 0.064 | 0.673 ± 0.096 | 0.800 ± 0.060 | 0.678 ± 0.131 | **0.652 ± 0.024** | 0.422 ± 0.105 |
| HSV | 0.733 ± 0.031 | 0.713 ± 0.044 | 0.635 ± 0.055 | 0.701 ± 0.045 | 0.744 ± 0.029 | 0.739 ± 0.055 | 0.839 ± 0.012 | 0.749 ± 0.042 | 0.618 ± 0.064 | 0.460 ± 0.022 |
| STAUG-S | **0.782 ± 0.042** | **0.775 ± 0.090** | **0.689 ± 0.060** | **0.715 ± 0.042** | 0.740 ± 0.050 | 0.747 ± 0.082 | **0.859 ± 0.049** | 0.721 ± 0.081 | 0.567 ± 0.064 | 0.441 ± 0.052 |
| RandStainNA | 0.694 ± 0.019 | 0.724 ± 0.022 | 0.655 ± 0.026 | 0.667 ± 0.024 | 0.715 ± 0.021 | **0.754 ± 0.012** | 0.799 ± 0.030 | **0.781 ± 0.022** | 0.507 ± 0.066 | 0.460 ± 0.051 |
| STAUG-L | 0.737 ± 0.032 | 0.723 ± 0.035 | 0.649 ± 0.051 | 0.680 ± 0.041 | 0.694 ± 0.090 | 0.672 ± 0.114 | 0.819 ± 0.029 | 0.676 ± 0.076 | 0.552 ± 0.075 | 0.406 ± 0.111 |
| ISW-P | 0.748 ± 0.047 | 0.673 ± 0.100 | 0.609 ± 0.135 | 0.661 ± 0.072 | 0.691 ± 0.059 | 0.640 ± 0.109 | 0.785 ± 0.053 | 0.578 ± 0.136 | 0.564 ± 0.105 | 0.378 ± 0.086 |
| ISW-STAUG | 0.734 ± 0.030 | 0.683 ± 0.051 | 0.624 ± 0.043 | 0.652 ± 0.039 | 0.655 ± 0.063 | 0.621 ± 0.078 | 0.754 ± 0.053 | 0.542 ± 0.119 | 0.534 ± 0.049 | 0.370 ± 0.062 |
| Proposed | 0.733 ± 0.026 | 0.708 ± 0.048 | 0.618 ± 0.039 | 0.672 ± 0.039 | 0.733 ± 0.015 | 0.722 ± 0.022 | 0.809 ± 0.035 | 0.693 ± 0.072 | 0.648 ± 0.109 | **0.478 ± 0.056** |
| Proposed + STAUG-S | 0.767 ± 0.028 | 0.715 ± 0.021 | 0.620 ± 0.023 | 0.698 ± 0.020 | **0.748 ± 0.032** | 0.733 ± 0.033 | 0.829 ± 0.014 | 0.682 ± 0.100 | 0.592 ± 0.055 | 0.456 ± 0.049 |

## C.2. Organ-wise scores

Table 7: Unnormalized Dice scores for each tissue type on the public and the private test sets of the HPA+HuBMAP2022 dataset. STINV: Stain-invariant training. CA: Channel attention.

| Method | Public Test HPA + HuBMAP | | | | | Private Test HuBMAP | | | | |
|---|---|---|---|---|---|---|---|---|---|---|
| | Kidney | Prostate | Large Intestine | Spleen | Lung | Kidney | Prostate | Large Intestine | Spleen | Lung |
| ResNet-50 | | | | | | | | | | |
| Baseline | 0.072 ± 0.006 | 0.104 ± 0.038 | 0.089 ± 0.003 | 0.135 ± 0.026 | 0.078 ± 0.008 | 0.012 ± 0.008 | 0.073 ± 0.054 | 0.060 ± 0.002 | 0.121 ± 0.025 | 0.132 ± 0.008 |
| STINV | 0.084 ± 0.012 | **0.122 ± 0.021** | 0.091 ± 0.005 | **0.144 ± 0.014** | **0.087 ± 0.003** | 0.028 ± 0.016 | **0.095 ± 0.027** | 0.061 ± 0.006 | **0.128 ± 0.014** | **0.143 ± 0.004** |
| STINV + CA | **0.099 ± 0.004** | 0.119 ± 0.049 | **0.095 ± 0.011** | 0.130 ± 0.009 | 0.083 ± 0.002 | **0.056 ± 0.007** | 0.086 ± 0.061 | **0.064 ± 0.013** | 0.111 ± 0.015 | 0.136 ± 0.003 |
| ConvNext-Tiny | | | | | | | | | | |
| Baseline | 0.101 ± 0.020 | 0.135 ± 0.006 | **0.106 ± 0.002** | **0.180 ± 0.001** | 0.063 ± 0.000 | 0.051 ± 0.025 | 0.114 ± 0.010 | **0.078 ± 0.002** | **0.154 ± 0.001** | 0.114 ± 0.000 |
| STINV + CA | **0.110 ± 0.012** | **0.151 ± 0.007** | 0.102 ± 0.001 | 0.173 ± 0.010 | 0.063 ± 0.000 | **0.065 ± 0.015** | **0.132 ± 0.007** | 0.074 ± 0.001 | 0.147 ± 0.014 | 0.114 ± 0.000 |

For ResNet-50, notable improvements can be observed for kidney images when both stain invariant training and channel attention (CA) are introduced (Table 7). Results slightly fluctuate for the prostate, lung, and large intestine when CA is used. CA negatively affects segmentation performance on spleen samples. One of the possible explanations is that

model might rely on some of the suppressed features for detecting white pulp in the spleen. Or it might require longer training because it has a more complex appearance compared to tissues in other organs. For ConvNext-Tiny, there is an increase in scores for kidney, and prostate samples, while large intestine and lung segmentation results remain stable. In the case of the spleen, a marginal decrease can be observed. This might come from the stochasticity of the training process or the same reasons outlined for ResNet-50 results on the spleen.

## C.3. Qualitative Analysis

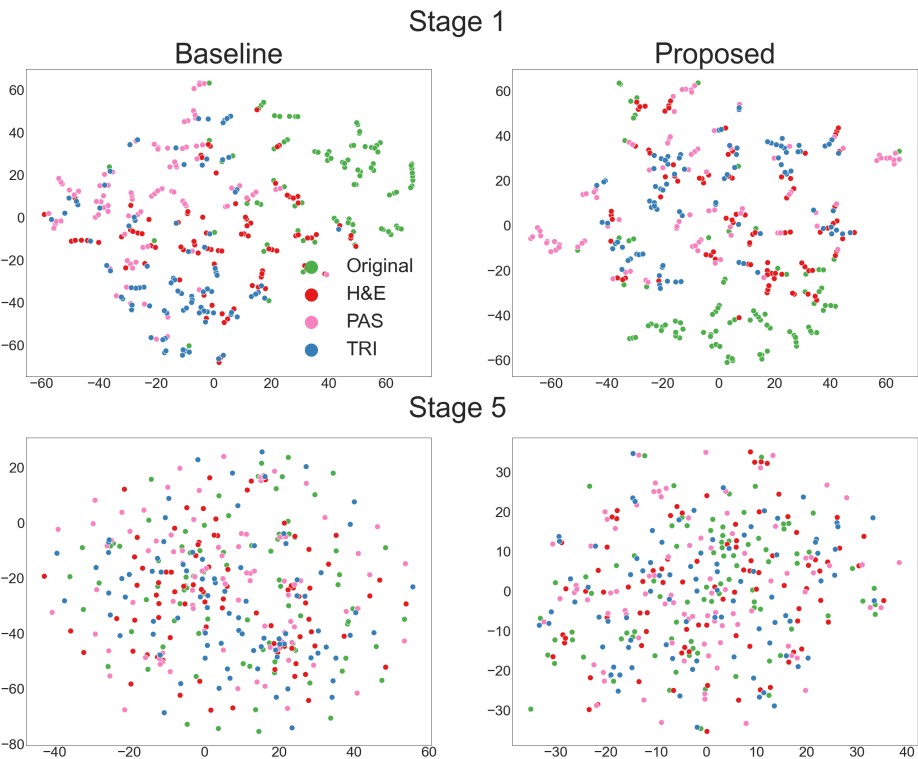

Figure 9: t-SNE visualization of learned representations (ResNet-50).

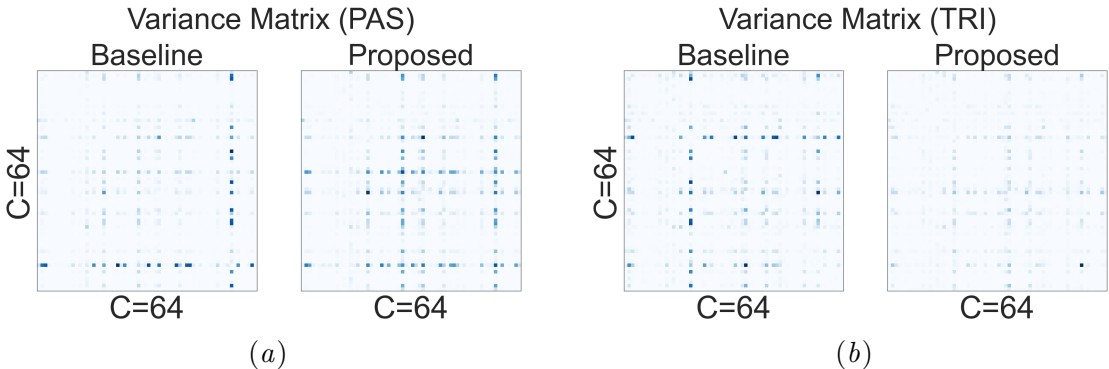

Figure 10: PAS, TRI variance matrices for synthesized validation set.

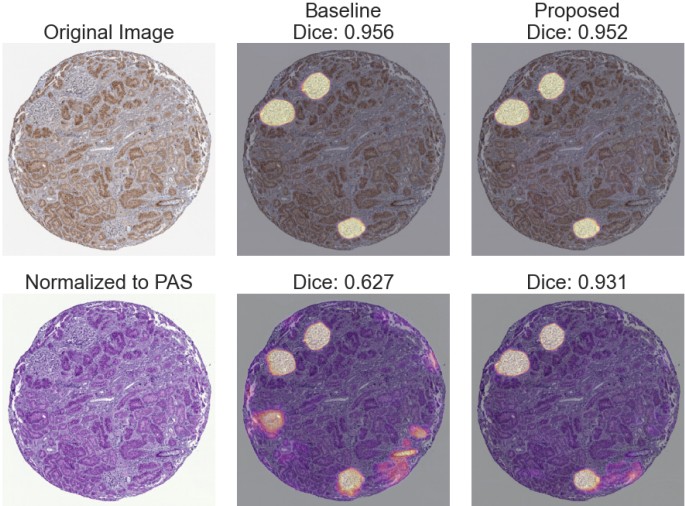

Figure 11: Predictions from baseline and modified model with our method for a kidney (glomerulus) image and its stain normalized version (PAS) from HPA+HuBMAP 2022 dataset.

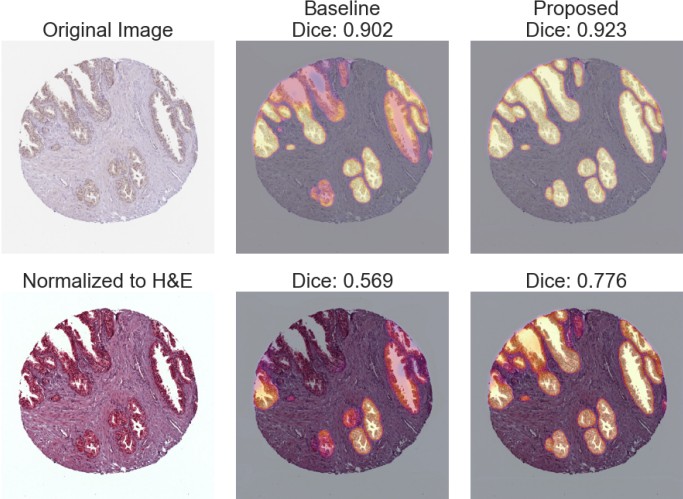

Figure 12: Predictions from baseline and modified model with our method for a prostate (glandular acinus) image and its stain normalized version (H&E) from HPA+HuBMAP 2022 dataset.

## Appendix D. Additional Experiments

### D.1. Positioning and Choice of Downsampling

We investigate how using the proposed method after later encoder stages affect overall results with ResNet-50 (Table 8). Findings suggest that there is a negative correlation between using deeper layer outputs and generalizability for the segmentation task. One possible explanation may come from the higher complexity of features of deeper stages. In addition, gradient reversal can cause excess weight disturbance during training, leading to insufficient learning of semantically meaningful representations.

Additionally, we experiment with different design choices, such as methods for downsampling of intermediate outputs (Table 9). For our training set, maxpooling appears to be a more suitable choice. Overall, we view this architectural choice as a hyperparameter that can be adapted based on the main model and training setup.

### D.2. Effect of Channel Attention

We observe improvements in segmentation performance compared to the baseline when stain-invariant training is introduced (Table 10). Though emphasizing domain-specific features through channel attention does not further increase public and private test scores (HPA+HuBMAP 2022), it significantly improves results on other test datasets. We also investigate the combination of stain-adversarial training and stain augmentation with and without attention branch. We find that the former combination shows the best results.

Table 8: How dice scores change when the proposed method is used after different encoder (ResNet-50) stages.

| Encoder Stage | Validation: HPA | | | | Public Test HPA+HuBMAP | Private Test HuBMAP | NEPTUNE | | | | HuBMAP21 | AIDPATH |
|---|---|---|---|---|---|---|---|---|---|---|---|---|
| | - | H&E | PAS | TRI | | H&E + PAS | H&E | PAS | TRI | SIL | PAS | PAS |
| w\o | 0.691 ± 0.044 | 0.605 ± 0.053 | 0.509 ± 0.066 | 0.493 ± 0.098 | 0.477 ± 0.064 | 0.399 ± 0.079 | 0.490 ± 0.036 | 0.542 ± 0.028 | 0.518 ± 0.048 | 0.386 ± 0.030 | 0.249 ± 0.075 | 0.363 ± 0.039 |
| $5^{th}$ | 0.670 ± 0.027 | 0.502 ± 0.082 | 0.338 ± 0.123 | 0.433 ± 0.033 | 0.452 ± 0.074 | 0.360 ± 0.094 | 0.330 ± 0.046 | 0.344 ± 0.044 | 0.364 ± 0.041 | 0.192 ± 0.019 | 0.145 ± 0.020 | 0.239 ± 0.086 |
| $4^{th}$ | 0.638 ± 0.027 | 0.427 ± 0.057 | 0.367 ± 0.082 | 0.353 ± 0.020 | 0.487 ± 0.083 | 0.383 ± 0.078 | 0.263 ± 0.035 | 0.356 ± 0.020 | 0.298 ± 0.061 | 0.202 ± 0.051 | 0.201 ± 0.023 | 0.233 ± 0.021 |
| $3^{rd}$ | 0.691 ± 0.031 | 0.601 ± 0.021 | 0.544 ± 0.055 | 0.494 ± 0.025 | 0.500 ± 0.024 | 0.424 ± 0.022 | 0.497 ± 0.099 | 0.543 ± 0.091 | 0.492 ± 0.107 | 0.397 ± 0.085 | 0.261 ± 0.024 | 0.375 ± 0.062 |
| $2^{nd}$ | 0.719 ± 0.007 | 0.595 ± 0.027 | 0.501 ± 0.075 | 0.525 ± 0.033 | **0.578 ± 0.015** | **0.504 ± 0.018** | 0.520 ± 0.136 | 0.571 ± 0.041 | 0.538 ± 0.085 | 0.424 ± 0.067 | 0.289 ± 0.058 | 0.361 ± 0.058 |
| $1^{st}$ | **0.721 ± 0.025** | **0.631 ± 0.018** | **0.567 ± 0.014** | **0.568 ± 0.071** | 0.526 ± 0.059 | 0.453 ± 0.073 | **0.593 ± 0.025** | **0.637 ± 0.057** | **0.566 ± 0.130** | **0.501 ± 0.115** | **0.295 ± 0.036** | **0.476 ± 0.030** |

Table 9: How using different downsampling methods affects results (Dice score) of the proposed approach. AVG: average pooling. SCONV: sequential convolutional layers. MAX: max pooling.

| Method | Validation: HPA | | | | Public Test HPA+HuBMAP | Private Test HuBMAP | NEPTUNE | | | | HuBMAP21 | AIDPATH |
|---|---|---|---|---|---|---|---|---|---|---|---|---|
| | - | H&E | PAS | TRI | | H&E + PAS | H&E | PAS | TRI | SIL | PAS | PAS |
| Baseline | 0.691 ± 0.044 | 0.605 ± 0.053 | 0.509 ± 0.066 | 0.493 ± 0.098 | 0.477 ± 0.064 | 0.399 ± 0.079 | 0.490 ± 0.036 | 0.542 ± 0.028 | 0.518 ± 0.048 | 0.386 ± 0.030 | 0.249 ± 0.075 | 0.363 ± 0.039 |
| AVG | 0.714 ± 0.027 | **0.652 ± 0.020** | 0.506 ± 0.025 | 0.547 ± 0.015 | 0.512 ± 0.054 | 0.434 ± 0.065 | 0.414 ± 0.041 | 0.512 ± 0.034 | 0.543 ± 0.027 | 0.402 ± 0.067 | 0.244 ± 0.021 | 0.343 ± 0.037 |
| SCONV | **0.740 ± 0.003** | 0.634 ± 0.023 | 0.528 ± 0.083 | 0.544 ± 0.062 | **0.578 ± 0.042** | **0.514 ± 0.052** | 0.530 ± 0.096 | 0.586 ± 0.077 | 0.478 ± 0.082 | 0.462 ± 0.109 | **0.297 ± 0.010** | 0.461 ± 0.025 |
| MAX | 0.721 ± 0.025 | 0.631 ± 0.018 | **0.567 ± 0.014** | **0.568 ± 0.071** | 0.526 ± 0.059 | 0.453 ± 0.073 | **0.593 ± 0.025** | **0.637 ± 0.057** | **0.566 ± 0.130** | **0.501 ± 0.115** | 0.295 ± 0.036 | **0.476 ± 0.030** |

Table 10: Effect of adding channel attention. Encoder: ResNet-50. STINV: only stain invariant branch. CA: channel attention based on detecting sensitive covariances. STAUG: stain augmentation.

| Method | Validation: HPA | | | | Public Test HPA+HuBMAP | Private Test HuBMAP | NEPTUNE | | | | HuBMAP21 | AIDPATH |
|---|---|---|---|---|---|---|---|---|---|---|---|---|
| | - | H&E | PAS | TRI | | H&E + PAS | H&E | PAS | TRI | SIL | PAS | PAS |
| Baseline | 0.691 ± 0.044 | 0.605 ± 0.053 | 0.509 ± 0.066 | 0.493 ± 0.098 | 0.477 ± 0.064 | 0.399 ± 0.079 | 0.490 ± 0.036 | 0.542 ± 0.028 | 0.518 ± 0.048 | 0.386 ± 0.030 | 0.249 ± 0.075 | 0.363 ± 0.039 |
| STINV | **0.731 ± 0.012** | 0.651 ± 0.020 | 0.581 ± 0.037 | 0.529 ± 0.039 | 0.528 ± 0.045 | 0.455 ± 0.054 | 0.469 ± 0.078 | 0.574 ± 0.097 | 0.550 ± 0.089 | 0.419 ± 0.098 | 0.269 ± 0.037 | 0.371 ± 0.031 |
| STINV + CA | 0.721 ± 0.025 | 0.631 ± 0.018 | 0.567 ± 0.014 | 0.568 ± 0.071 | 0.526 ± 0.059 | 0.453 ± 0.073 | 0.593 ± 0.025 | 0.637 ± 0.057 | 0.566 ± 0.130 | 0.501 ± 0.115 | 0.295 ± 0.036 | 0.476 ± 0.030 |
| STINV + STAUG | 0.729 ± 0.006 | 0.685 ± 0.009 | 0.602 ± 0.045 | **0.668 ± 0.014** | 0.599 ± 0.022 | 0.536 ± 0.020 | 0.707 ± 0.083 | 0.728 ± 0.056 | 0.759 ± 0.029 | 0.566 ± 0.043 | 0.383 ± 0.034 | 0.467 ± 0.071 |
| STINV + STAUG + CA | 0.711 ± 0.020 | **0.693 ± 0.025** | **0.616 ± 0.023** | 0.650 ± 0.025 | **0.622 ± 0.029** | **0.567 ± 0.029** | **0.736 ± 0.060** | **0.786 ± 0.028** | **0.808 ± 0.023** | **0.733 ± 0.040** | **0.483 ± 0.018** | **0.470 ± 0.051** |

