# OpenReview forum: "Improving Stain Invariance of CNNs for Segmentation by Fusing Channel Attention and Domain-Adversarial Training"
_MIDL.io/2023/Conference — MIDL 2023 Poster_

### Official Review · Reviewer_ndGP · 2023-02-05

**Confidence:** 3
**Preliminary Rating:** 4
**Recommendation:** Poster

**Summary:**

This paper aims to address the stain variance in WSIs, to train a segmentation model with higher generalization capability. The main idea is to detect the feature channels that are stain sensitive using the correlation between original feature maps and the corresponding stained augmented feature map in the shallow layers of the segmentation model. Then domain-adversarial training using optimal stain vectors is imposed on the detected channels (obtained by attention operation). The proposed method shows promising performance on four datasets.

**Strengths:**

* Well written and easy to follow.
* The proposed method is elegant and easy to implement.
* Promising performance was achieved on four datasets, ensuring the generalization.
* The results are convincing.


**Weaknesses:**

* The notation should be improved. For example, matrix or vector should be written in bold.
* The explanation of ‘optimal stain vectors’ is not given, which makes it hard to understand the motivation of the RMSE loss L_s. Please note that being self-contained is important to the readability of a paper.
* Authors are suggested to provide the computation efficiency in training compared with other methods.


**Deanonymize Review:**

no

**Paper Type:**

methodological development

**Questions To Address In The Rebuttal:**

* Could the authors explain more in detail about the optimal stain vectors, please?
* How about the training efficiency of the proposed method in comparison with other methods?
* The FC layer upon the V matrix is learnable. How can it be guaranteed the output attention scores are exactly corresponding to the stain sensitive channels? This question maybe related to the RMSE loss.

---

### Official Review · Reviewer_wYEF · 2023-02-05

**Confidence:** 4
**Preliminary Rating:** 3

**Summary:**

The paper presents an approach based on channel attention to handle the stain variance for the semantic segmentation under  multi-center, multi-stain settings. The proposed methodology also complements the existing stain augmentation techniques. The results are presented on four datasets, and proposed methodology has provided gains on some of them.

**Strengths:**

The paper is easy to follow, and results have presented extensively on four datasets with two different backbones to fully analyze the scope of the proposed methodology. The authors have also followed the standard practice of reporting the mean and std over the multiple iterations. Further, author have presented a deep analysis of the different aspects of the methodology rather than presenting a heuristic approach. The approach generalizable, and can be incorporated in the existing architectures and can also be used with existing augmentation techniques.

**Weaknesses:**

1. The method is not performing consistently well for all the datasets as compared to other methods with both baselines (Table 1&2).  Further, method seems to effective only with STAUG-S, which shows its high sensitivity to the augmentation characteristics.
2. Although authors have shown comparison with different stain augmentation methods, the comparison should also be provided with related domain adaption methods such as https://openreview.net/forum?id=uXl3bZLkr3c

**Deanonymize Review:**

no

**Detailed Comments:**

There are some typos: S ∈ R^3x2, it should be S ∈ R3X2?

**Paper Type:**

methodological development

**Questions To Address In The Rebuttal:**

1. The paper should contain the sensitivity analysis to the augmentation degrees (strength). The comparison with some domain adaption methods should also be included.
2. Have authors also considered the alternate (modification) of Eq.3. ?

---

### Official Review · Reviewer_XS9o · 2023-02-05

**Confidence:** 4
**Preliminary Rating:** 5
**Recommendation:** Oral

**Summary:**

This paper proposes a new method to reduce domain shift to to staining variation for segmentation methods for histopathological images. The method does not require access to images from the new domain. Instead, it uses stain augmentation in the source domain to find features that are less sensitive to staining variation and use this to reweigh feature maps.

**Strengths:**

- Relevant problem
- Interesting methodological idea, and well carried out
- Paper contains a lot of useful (side-)comments, I think researchers in this field can learn a lot from this paper and I found it the most interesting paper from the batch I had to review
- Extensive comparison with other methods, solid experimental work


**Weaknesses:**

- Description data unclear (see below)
- Not clear if all data is public, major weakness if not, please make it possible for people to easily download all data to repeat all the experiments
- Code is kept secret, it should be shared with a permissive license, I encourage the authors to share it with reviewers in the rebuttal phase. Research from Umeå University showed (https://arxiv.org/abs/2210.11146) that (MIDL) papers regularly promise to share code but ultimately the code is not shared or the code shared is not of high-quality. In my opinion, code should be shared with reviewers during the review period and conferences should require that code is really shared.
- Paper writing can be improved (see below)

**Deanonymize Review:**

yes

**Detailed Comments:**

I would rewrite the abstract, it raises a lot of questions, eg "Our approach uses a channel attention mechanism that detects stain-specific features and a modified stain-invariant training scheme based on recent findings", what recent findings are meant? There are many such sentences, most are clear to me after reading the paper, but that’s not the idea of an abstract.

Writing style in general can be improved at many places. Some examples:
And there is no access to a test set —> add: from a different source
Escalating number —> growing number
Some are tissue extraction —> poor English, I suggest to use a language model to improve the English
such modification synthesizes —> idem

The data set description is hard to follow. Maybe a table is clearer. Describe where the data can be downloaded and what the license is, provide urls/references. It is not not clear to me if this data is public, if not, the work is not reproducible. Mention exactly what data you use or put it in a new repository and provide url, statements like ‘A subset of’ are confusing, in this way we do not know which slides were used in this work.

I suggest merging table 1 and 2, it is much easier to compare the results if the numbers are right below each other on the same page.

In the discussion it would be interesting to read about whether this technique should always be used, or can also be counterproductive, and if so, how do you know, not having access to external test data, if you should apply it?

What would you do if you do have (a small number of) external test images. How to take advantage of that if you want to use the method you propose? Not using them at all seems wasteful.

Would the method work with visual transformer methods? Could you comment on that?



**Paper Type:**

methodological development

**Questions To Address In The Rebuttal:**

See comments above.
-------------------------------------------------------------------------------------------------------------------------------------------------------------------------------------------

---

### Official Review · Reviewer_VtcG · 2023-02-07

**Confidence:** 4
**Preliminary Rating:** 1

**Summary:**

The authors propose a novel method to improve domain generalization in histopathology. The authors propose to apply stain augmentation on the source domain to find feature maps that are less sensitive to staining variation, hypothesizing that stain information is encoded as feature covariances. Domain-adversarial training using optimal stain vectors is then applied to the obtained by attention operation channels.

**Strengths:**

1. The authors tackle a relevant issue in computational pathology.

2. The authors propose a novel method for improving domain generalization in histopathology.

3. The authors report results on multiple datasets.


**Weaknesses:**

1. The study setup is flawed. The authors propose a solution to improve domain generalization in histopathology, however, none of the experiments, datasets choice, and baselines choice are made to demonstrate the perks of using this method to improve domain generalization.


2. Benchmarking against state-of-the-art is not present.

2.1. Even though the authors report results on 4 datasets in their study, none of these datasets is commonly used in literature as a benchmark for domain generalization. There are no reports of these datasets originating from various centers. Reporting results on common benchmarks such as Camelyon17 or MIDOG, where data from EXTERNAL centers are used for testing would allow to understand the contribution of this paper to the community.

2.2. Instead of comparing to state-of-the-art benchmarks from the literature, the authors compare results to a few self-defined baselines like random perturbations in HSV space, etc . The authors provide no clear justification for why the baselines were selected. It would be useful to see the comparison to at least one of the state-of-the-art methods reported in the literature.





**Deanonymize Review:**

no

**Detailed Comments:**

The Abstract lacks a description of the methodology.

**Paper Type:**

methodological development

**Questions To Address In The Rebuttal:**

1. Provide results on a public domain generalization benchmark dataset in histopathology (Camelyon17, MIDOG, TUPAC16,..). During the experiments ensure that the model is trained on data from a single institution and tested on data from external centers.

2. Compare to at least one of the state-of-the-art domain generalization methods from the literature[1,2,3].

3. The proposed method has methodological similarities with [2], it would be useful to see an ablation study on how the modifications proposed by the authors impact performance and domain generalization.

4. Provide a precise description of data used in this study, including centers of origin, scanners, and details of development/test center-wise data split.

[1.] D. Tellez et al., Quantifying the effects of data augmentation and stain color normalization in convolutional neural networks for computational pathology

[2.] M. Lafarge et al., Domain-Adversarial Neural Networks to Address the Appearance Variability of Histopathology Images

[3.] K. Faryna et al., Tailoring automated data augmentation to H&E-stained histopathology

---

### Meta-Review · Area_Chair_mrh2 · 2023-02-25

**Recommendation:** Accept (Poster)
**Confidence:** 3

**Metareview:**

The authors propose a method to improve generalizability of CNNs to different stains for whole slide images (WSI) by employing a channel attention mechanism and stain-adverarial training. The method is tested on multi-center, multi-stain datasets and compared to multiple other methods.

Reviewers agreed on the importance of the problem, the novelty and interest of the proposed approach, and the strength of extensive experiments on 4 datasets. However, there were extremely divergent overall ratings, with one claiming the paper was the most interesting of their batch, and another on the other end. The greatest concerns are regarding the experimental setup, where multiple reviewers note that there are missing comparisons for domain adaptation/generalization baselines, and rather experiments focus on comparison to different data augmentation approaches.

While the missing comparison to potential domain generalization baselines is a strong, valid concern, I feel the general interest of the approach, strength of evaluation using multiple public datasets and multiple stains, and comparison to state-of-the-art augmentation methods for stain generalizability, overall outweigh the noted experimental concerns.